# Colour segmentation of printed fabrics by integrating adaptive neural network and density peak clustering algorithm

Niu Meng[1,2]*

1 Department of Arts and Design, Shanghai Business School, Shanghai, China, 2 Shanghai Jiao Tong University, Shanghai, China

* hellonm2023@126.com

## Abstract

With the development of computer vision and image processing technology, color segmentation of printed fabrics has gradually become a key task in the textile industry. However, the existing methods often face the problems of low segmentation accuracy and poor computational efficiency when dealing with high complexity patterns and similar colors. To address the above problems, a new color segmentation algorithm for printed fabrics is proposed by integrating the self-organizing mapping network (SOM) in adaptive neural network and the density peak clustering algorithm. The method achieves topological mapping learning of color features through SOM, and then uses DPC for density-driven fine clustering division, which effectively improves the accuracy and stability of color segmentation. The experimental results show that the proposed method shortens the execution time by nearly 40% compared with the self-organized mapping network, and the average color difference (ΔE) of each region after color segmentation is as low as 0.7, which is significantly better than other algorithms. Meanwhile, in the detection of the four types of printed fabric samples, the obtained average color value is up to 87.49 (the higher the 0–100 score value indicates that the color is more significant), and the smallest standard deviation is 2.18 (the smaller the value indicates that the color segmentation is more centralized), which further verifies the comprehensive advantages of the algorithm in terms of segmentation accuracy and stability. In conclusion, the proposed method provides an effective reference for improving the quality and efficiency of color segmentation of printed fabrics.

## 1. Introduction

Printed fabric color segmentation is the process of dividing and extracting different color regions in printed fabrics, which is important for the design, production and sales of printed fabrics [1]. However, traditional image segmentation algorithms face a

**Data availability statement:** The data supporting the findings of this study are available within the article.

**Funding:** Ministry of Education Humanities and Social Sciences Research Youth Fund Project: A Study of the Military Attire Formations in Southern Inspection Paintings of the Ming and Qing Dynasties.

**Competing interests:** The authors have declared that no competing interests exist.

**Abbreviations:** ANN, Adaptive Neural Network; DPC, Density Peaks Clustering; SOM, Self-Organizing Map; ED, Euclidean distance; SC, Silhouette Coefficient.

series of challenges in printed fabric color segmentation due to the complexity of color distribution and the diversity of changes in printed fabrics. In view of this, scholars and researchers at home and abroad have carried out different degrees of research on color segmentation of printed fabrics [2]. For example, foreign scholars have proposed segmentation methods using machine learning, such as vector machines and convolutional neural networks [3,4]. These methods can realize the color segmentation of printed fabrics to a certain extent, but there are still some problems. For example, the processing effect of complex images is poor, and a large amount of computational resources are required. In recent years, adaptive neural networks have gradually become an important tool for unsupervised modeling in image segmentation by virtue of their superior ability in self-organization and topology preservation of high-dimensional data, among which Self-Organizing Map (SOM) has been widely used in color clustering and spatial structure reconstruction. Meanwhile, the Density Peaks Clustering (DPC) algorithm shows unique advantages in dealing with image regions with significant density variations by virtue of its ability to recognize cluster structures of arbitrary shapes and its flexibility in not needing to preset the number of clusters. Although both algorithms have their own characteristics, SOM suffers from insufficient boundary refinement and sensitivity to local structure distribution, while DPC faces the challenges of easy centroid misalignment and noise misjudgment in high-dimensional space. Based on this, the study proposes a color segmentation algorithm for printed fabrics that integrates SOM and DPC, aiming to further improve the image segmentation effect of printed fabrics. Specifically, on the one hand, the adaptive mapping and region delineation of high-dimensional color features are accomplished by the topology preservation ability of self-organizing mapping network in unsupervised learning, and on the other hand, the density peak clustering is used to enhance the density and cluster center determination of the initial results of the SOM, so as to improve the differentiation ability of the complex color structure and the boundary integrity expression. The contribution of the study is that a color segmentation model with both global mapping and local aggregation is proposed, which systematically solves the problems of poor identification of color proximity regions, weak expression of pattern boundaries and strong dependence on parameter tuning in the traditional methods, and provides theoretical support and methodological basis for improving the stability, scalability and industrial practicability of image segmentation of complex textured fabrics. This research is divided into five parts. The first part is a brief introduction to the overall content of the article, the second part is an analysis and summary of others' research, the third part describes how the model combining the SOM network and the DPC algorithm is constructed, the fourth part is a test of the model's performance. The last part is a summary of the article.

## 2. Related works

Printed fabric color segmentation is a challenging task, which is of great significance for printed fabric design and production. Traditional color segmentation methods for printed fabrics mainly rely on manual annotation and expert experience, which have the problems of high subjectivity and low efficiency. Therefore, researchers have begun to

explore new methods based on machine learning and clustering algorithms to solve this problem. In recent years, adaptive neural networks have gradually come into the public eye. This network can automatically adjust its own weights and structure according to the statistical laws of the input data. Son et al. [5] used ANN for piezoelectric ceramic actuator control, which significantly improves tracking accuracy and stability, but its application scope is concentrated in the dynamic control field. Chen et al. [6] constructed an ANN-based trajectory control model for the speed constraint problem of mobile robot, which embodies a good nonlinear modeling capability. Xue et al. [7] proposed an adaptive gradient descent mechanism to optimize the convergence performance of the ANN in a fully connected networks to optimize the convergence performance of ANN in fully connected networks. Tan et al. [8] introduced adaptive quantization control to resist network attacks and achieve multiplexed neural synchronization in complex network environments. Although the above studies demonstrated the adaptability and control advantages of ANN, they lacked a structure matching mechanism for the color segmentation task of complex texture images, which made it difficult to be directly migrated to high-density patterned fabric image processing scenarios.

Density peak clustering algorithm is a commonly used unsupervised clustering algorithm, which can be classified into different clusters according to the density distribution characteristics among samples. The shape between clusters is unrestricted and less sensitive to parameters. Meanwhile DPC can be applied to data with arbitrary density distribution, Therefore, it is mostly applied in the pattern recognition, image analysis and information extraction. Shi et al. [9] combined DPC with a probabilistic neural network for analog circuit fault diagnosis to improve classification accuracy while reducing the number of neurons. Guan et al. [10] optimized the hierarchical clustering efficiency of DPC on large-scale datasets through a correlation transfer mechanism. While Li et al. [11] combined DPC with an RBF network for mixed-data classification and achieved a high accuracy of 97.52%. In addition, Zheng et al. [12] optimized DPC based on k-nearest neighbor for industrial process monitoring and significantly improved detection stability. Although DPC shows good performance in the field of image analysis, its clustering results are sensitive to local density estimation and distance metrics, which makes it difficult to directly capture complex color structures in high-dimensional images. Some studies have attempted to integrate neural networks with clustering algorithms. For example, Gharehchopogh et al. [13] improved the image segmentation performance by improving the African vulture optimization algorithm, but its applicable images are mainly medical and natural images; Zhou et al. [14] improved the accuracy and speed of fabric defect detection based on the improved YOLOv5s model, but the target is more oriented to structural defect recognition; Sikka et al. [15] constructed a fabric detection model based on multilevel backpropagation network to construct a fabric detection model, which is difficult to cope with the problem of boundary extraction in the region of similar color despite its robustness. There is a lack of research on color adaptive recognition and high-resolution segmentation for high-density, multi-color fusion scenarios in printed fabrics.

In summary, although the combination of SOM and clustering methods has been applied in some literatures, such as realizing clustering in remote sensing images or texture images, most of these methods have not considered the consistency requirements of the high-dimensional spatial distribution of the color features in the printed fabrics with the expression of the physical chromatic aberration, and they have not integrated the advantages of the self-organized mapping of the neural network and the nonparametric clustering ability of the DPC. Therefore, this study focuses on the integration of SOM and DPC, and proposes a SOM-DPC color segmentation model for printed fabrics by introducing a clustering mechanism based on density distribution while maintaining the advantages of spatial topological mapping structure. The model not only improves the accuracy of color recognition in hyperspectral images, but also enhances the sensitivity to the adjacent color gamut demarcation, thus filling in the deficiencies of existing methods in the control of high-dimensional color representation and segmentation accuracy.

## 3. Methods

### 3.1. Design of color segmentation model for printed fabrics combining adaptive neural network and density peak clustering algorithm

Printed fabric color segmentation is an important research topic, which is significant for the design and production of printed fabric [16]. Traditional color segmentation methods for printed fabrics often rely on expert experience and manual

labeling, which have the problems of high subjectivity and low efficiency. Section I introduces a neural network model applied in the color segmentation method of printed fabrics, and Section II introduces a clustering model applied in the color segmentation method of printed fabrics. Section III combines the two to propose a new color segmentation model for printed fabrics.

### 3.1.1. Color segmentation model construction of printed fabric based on adaptive neural network.

ANN is a class of machine learning algorithms capable of extracting features and performing classification through learning. This class of algorithms is highly capable of learning and adaptive, and can improve accuracy with a large amount of sample data. In adaptive network classification, self-organizing mapping network is a special type of unsupervised learning neural network [17]. This neural network is adaptive and automatically organizes the topology of the input data through a learning process that maps the high-dimensional input data onto a low-dimensional grid structure to form a kind of "map". In the color segmentation of printed fabrics, SOM can extract the regions of different colors by learning the features of printed fabric images and perform classification and segmentation. The structure of SOM is shown in Fig 1.

As can be seen from Fig 1, the network has an input layer and an output layer, the input layer contains many input nodes, and there are n nodes in the input layer if the input data is of n dimensions. The output layer contains many output neurons, and the individual neurons are arranged in a specific structure in a quadrilateral or hexagonal manner. Overall all the input nodes and output neurons are interconnected and inseparable.

The output layer, also known as the competitive layer, has a total of five steps in its competitive learning process. Assuming that all pixel points in the hyperspectral image of the printed fabric is N, the similarity calculation of individual pixels is expressed in terms of Euclidean distance (ED) measure as shown in equation (1).

$$d_l = \|y_i, w_l\|, l = 1, 2, 3, M \tag{1}$$

In equation (1), $w_l$ denotes the normalized weight vector, $y_i$ denotes the input value, $d_l$ denotes the Euclidean distance, $M$ denotes the connection weight vectors, and $l$ stands for the splice weight vector. The normalized data is filtered to obtain the neuron with the smallest Euclidean distance as shown in equation (2).

$$c = \min d_l, l = 1, 2, 3, M \tag{2}$$

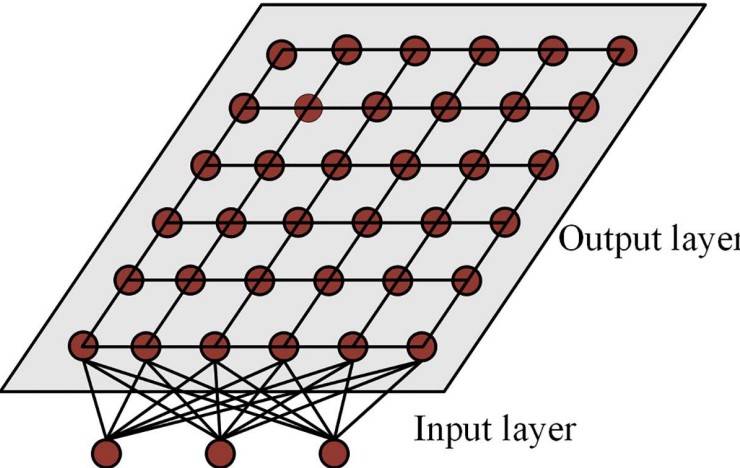

**Fig 1. Two-dimensional SOM neural network model diagram.**

In Eq. (2), *c* denotes the neuron with the smallest Euclidean distance, and the computation is performed on the obtained neuron with the smallest distance, then the radius of the neuron's field at this point is represented as shown in Eq. (3).

$$h(\rho, t) = \exp(-\frac{\rho^2}{2\sigma^2(t)})$$

(3)

In Eq. (3), $\rho$ stands for the position of the neuron on the output layer, and $\sigma(t)$ denotes the domain radius function, which decreases with the growth of learning time while the domain radius becomes smaller. Continue to update the connection weight vectors of the input nodes and all neurons in the domain, the update process is shown in equation (4).

$$w_p = w_p + \alpha(t)h(\rho, t)(x - w_p)$$

(4)

In Eq. (4), $\alpha(t)$ denotes the learning function, $w_p$ denotes the connection right vector after continuous updating, and *t* denotes the learning time. As the learning time is lengthened, the continuously updated domain function at this time is shown in equation (5).

$$\sigma(t) = \sigma_0 \exp(-\frac{t}{\lambda})$$

(5)

In Eq. (5), $\sigma(t)$ denotes the domain function, and $\lambda$ denotes the ratio between the iterations and the initial neighborhood value. After determining the updated neighborhood function, the learning rate at this point is expressed as shown in equation (6).

$$\alpha(t) = a_0 \exp(-\frac{t}{\lambda})$$

(6)

In Eq. (6), $\alpha_t$ denotes the learning rate that is constantly updated with the neighborhood function. $\lambda$ The meaning of the ratio definition of is shown in Eq. (7).

$$\lambda = N/\log(\sigma_0)$$

(7)

In Eq. (7), *N* denotes the maximum number of iterations. After continuous learning, learning stops when the rate decays to 0 or the maximum number of iterations is reached. The SOM competitive learning process is shown in Fig 2.

In Fig 2, the parameters are first initialized, and after completion, the normalized data are input and these data are compared with the similarity. After comparison the neuron with the smallest ED is selected and adjusted to update the weights, neighborhood function and learning rate. If learning has been completed at this point then time +1, if not then repeat the step of selecting the neuron with the smallest ED to improve the learning accuracy. At this point the maximum iterations are reached or the learning rate is reduced to 0, then the learning is ended.

**3.1.2. Color segmentation model construction of printed fabrics combined with DPC algorithm.** The DPC was proposed by Alex Rodriguez and Alessandro Laio in 2014 [18]. Since the algorithm mainly targets distance and density computation, it can detect clusters formed in non-oriented spaces, such as non-spherical and non-two-dimensional spatial types. The formula for calculating the local density in the DPC algorithm is shown in equation (8).

$$\rho_i = \sum_j \chi(d_{ij} - d_c)$$

(8)

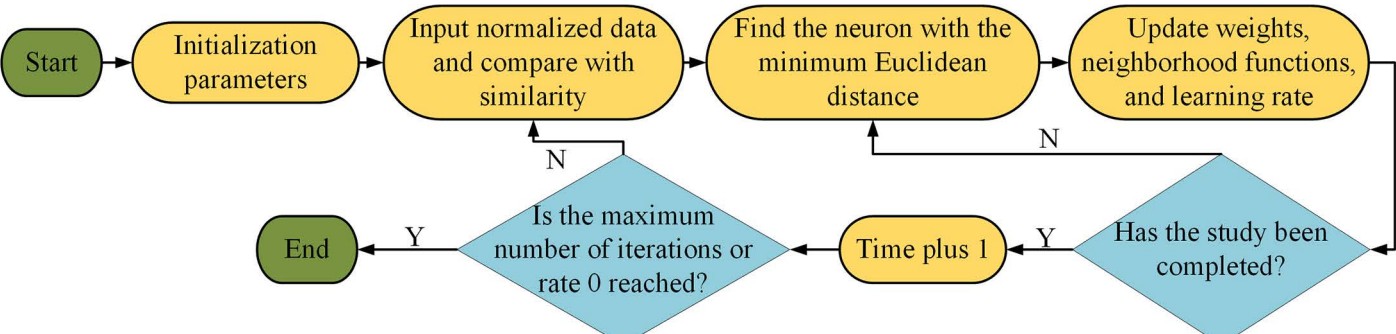

**Fig 2. SOM competes for the learning process.**

In Eq. (8), $\rho_i$ stands for the local density, $d_c$ stands for the truncation distance, and $d_{ij}$ stands for the ED between data point $i$ and data point $j$. The relative distance denotes the minimum ED between a data point and a data point with a larger localized density than that point, which is calculated as shown in Eq. (9).

$$\delta_i = \min(d_{ij}), \rho_i < \rho_j \tag{9}$$

In Eq. (9), $\sigma_i$ denotes the relative distance. In non-two-dimensional space, it is assumed that there exists a specific coordinate for each data point in the data set, and this way of representing the data points in non-two-dimensional space with specific coordinates is called distribution map, and the study takes the data set S as an example, and at this time the distribution map is shown in Fig 3.

From Fig 3, there is a data point $x$ in the data set S. The data point is the center of the data set S, and is the radius of the circle. At this time, the area after making a circle with this data point as the center and $dc$ as the radius is represented by a red circle. The data points in the circle represent the local density of the data point $x$. If a data point has a large local density, and at the same time the ED between the data point and the data point with greater density is also larger, then the data point can be called the center of the cluster. And the average distance from the data point to other data points in the cluster at this time is shown in equation (10).

$$a(x) = \frac{1}{n_i - 1} \sum d(x, y) \tag{10}$$

In Eq. (10), $a(x)$ denotes the average distance, $d(x, y)$ stands for the ED between data points $x$ and $y$, and $n_i$ denotes the number of data points in the $i$th cluster. If $a(x)$ is smaller, it means the density in the cluster is higher. In addition, the average distance between data point $x$ and data points in other clusters is calculated as shown in equation (11).

$$b(x) = \min\left[\frac{1}{n_j} \sum d(x, y)\right] \tag{11}$$

In Eq. (11), $n_j$ denotes the number of nodes in the first $j$ data point. $b(x)$ The larger the value, the less the data point $x$ belongs to other clusters. The contour coefficient of the data point $x$ is calculated as shown in Eq. (12) for intra-cluster calculation.

$$S(x) = \frac{b(x) - a(x)}{\max(b(x) - a(x))} \tag{12}$$

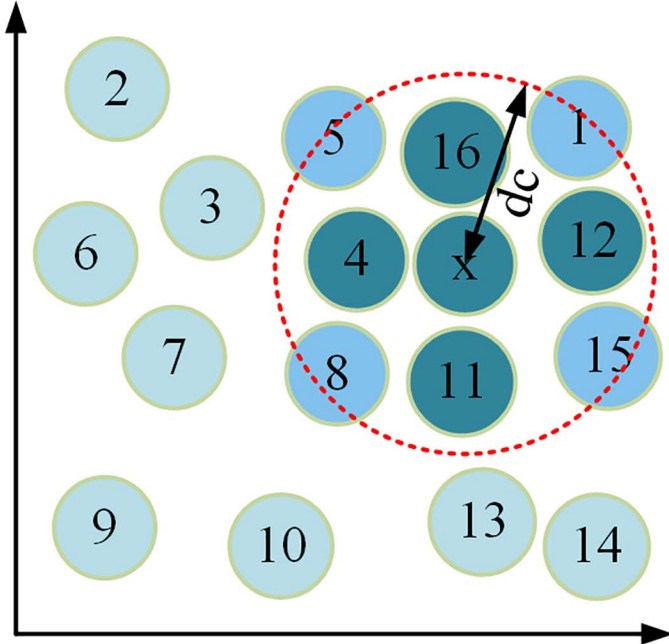

**Fig 3. Distribution of data set S.**

In Eq. (12), $b(x)$ denotes the inter-cluster separation, and $a(x)$ denotes the average intra-cluster distance and intra-cluster tightness. By averaging the silhouette coefficients of all points, the silhouette coefficient (Silhouette Coefficient, SC) of the whole cluster is obtained, as shown in Eq. (13).

$$SC = \frac{1}{n_c} \sum_{i=1}^{C_{k\,max}} \sum S(x)$$

(13)

In Eq. (13), $C_{k\,max}$ stands for the maximum clusters and $SC$ stands for the contour coefficient of the entire cluster. The contour coefficient combines the calculation of separation and tightness and is used to describe the degree of fitness within the entire cluster. Combining the above calculation process, the execution of the DPC algorithm is shown in Fig 4.

From Fig 4 DPC execution steps, firstly the matrix distance is calculated using the sample dataset and the neighborhood truncation distance is determined. Then local densities and relative distances are calculated and distribution and decision diagrams are drawn. Finally, the data points in the graph are screened to determine the clustering center points, and the data points that are not clustering centers are grouped. Distance detection can also be performed on the clustered center data points to affirm the calculation results.

**3.1.3. Color segmentation model construction of printed fabric combining SOM neural network and DPC algorithm.** In order to achieve the purpose of separating different color regions in the printed fabric image, the most important part is that the colors in the fabric image must be determined and the boundary of each color region must be determined. However, due to the different nature of the materials, the colors in printed fabrics usually do not have clear boundaries and cannot be directly observed by the naked eye [19]. Therefore, the research combines the previous theory of SOM neural network and the theory of DPC algorithm to propose a new color SOM-DPC automatic segmentation algorithm for printed fabrics. The neighborhood visualization of printed fabric hyperspectral image by this new algorithm is shown in Fig 5.

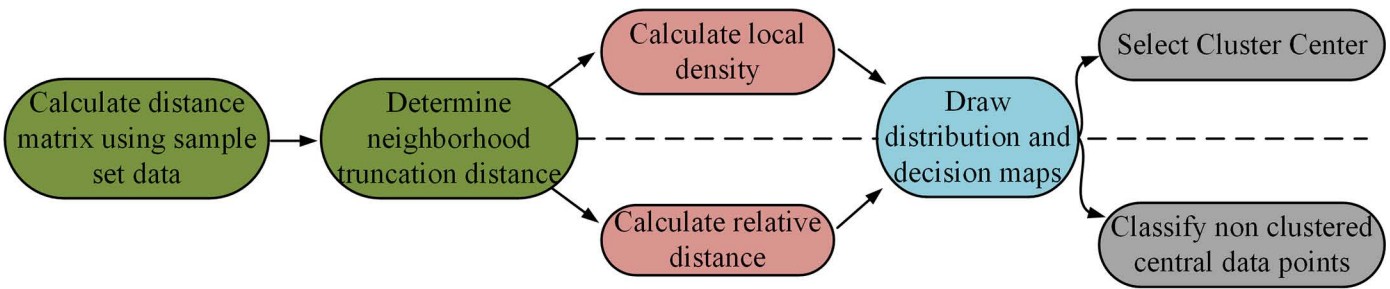

**Fig 4. The execution process of DPC algorithm.**

In Fig 5, the SOM-DPC can order the irregular hyperspectral images of printed fabrics. Each positive hexagon represents an output neuron of a competing layer. The darker regions indicate two neighboring neurons with a large difference between them, and the lighter regions are represented as two neighboring neurons with a smaller difference. In addition, each input color feature corresponds to a plane which represents the weights. After learning the competition, the data points where each neuron meets can then be calculated. If there are more data points, clusters can be formed, which represent the main colors in the hyperspectral image of the printed fabric. If there are fewer data points, the boundary of the data points represents the boundary of the cluster, i.e., the transition color between two colors. In addition, after summarizing the functions of the new algorithm, the study proposes the segmentation process of the new algorithm, as shown in Fig 6.

From Fig 6, the summarized SOM-DPC segmentation consists of five main steps. The first step acquires hyperspectral images of printed fabrics and converts the spectral reflectance corresponding to the pixel points in the region to be segmented into recognizable values. The second step divides and categorizes these data by SOM neural network. The third step is the secondary segmentation of the data by DOC algorithm. The fourth step is to determine the best color in the printed fabric based on the cluster validity index. The fifth step is to divide the entire printed fabric into sub-images with each color image as a unit.

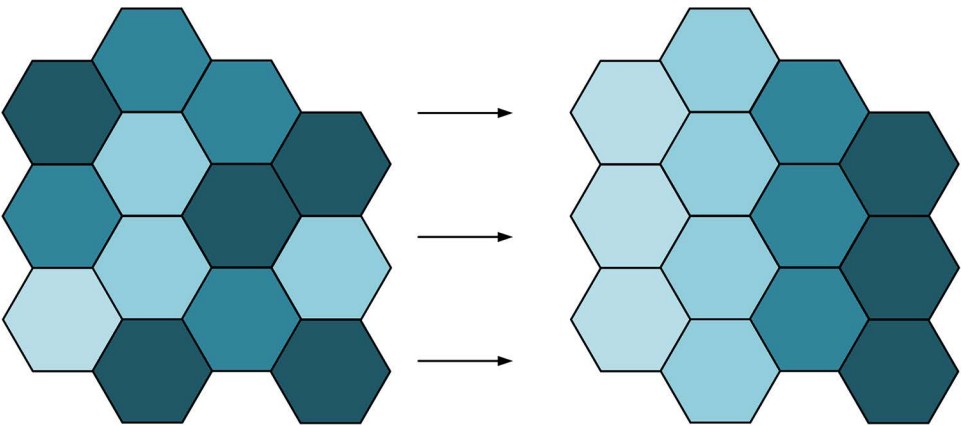

**Fig 5. The results of hyperspectral image neighborhood visualization by new algorithm.**

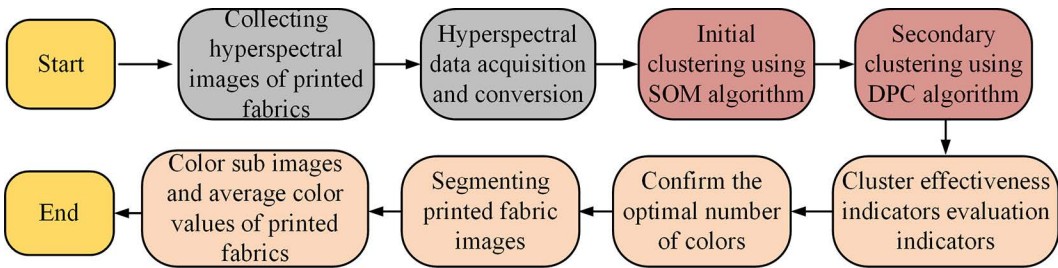

**Fig 6. The segmentation process of this new algorithm.**

## 4. Results

### 4.1. Analysis of experimental results of color segmentation model for printed fabrics combining adaptive neural networks and density peak clustering algorithm

To test the performance effect of the SOM-DPC, the study firstly takes a specific cluster analysis metric as a benchmark, such as Silhouette Coefficient (SC). The actual segmentation effect of the SOM-DPC algorithm on the color of printed fabrics is tested. Secondly, the study compares the SOM-DPC segmentation algorithm with several existing segmentation algorithms to investigate the performance of the algorithm in the same class of segmentation models.

**4.1.1. Assessment of indicators.** In order to systematically measure the effectiveness of the proposed SOM-DPC color segmentation algorithm for printed fabrics, the study employs four types of common evaluation metrics for performance evaluation, which are Color Difference, Silhouette Coefficient (SC), Execution Time, and Lab Accuracy metrics. Color difference is used to measure the color distance between different color regions, which is calculated based on CIE Lab color space and defined as shown in equation (14).

$$\Delta E = \sqrt{(L_1 - L_2)^2 + (a_1 - a_2)^2 + (b_1 - b_2)^2} \tag{14}$$

In Eq. (14), $L_1$, $a_1$, $b_1$ and $L_2$, $a_2$, $b_2$ represent the luminance (L), red-green axis (a) and yellow-blue axis (b) coordinate values of two color regions in Lab space, respectively. The smaller value of color difference represents the more accurate color segmentation and the more natural color transition of adjacent regions. The contour coefficient is used to measure the tightness of each data point in the clustering result with its cluster and the separation of other clusters, and its calculation formula is shown in equation (15).

$$SC(i) = \frac{b(i) - a(i)}{\max\{a(i), b(i)\}} \tag{15}$$

In Eq. (15) $a(i)$ denotes the average distance between sample point $i$ and other points within the same cluster; $b(i)$ denotes the average distance from sample point $i$ to the nearest neighboring cluster. SC value is between $[-1, 1]$, and the closer the value is to 1, the better the clustering effect is, and the clearer the segmentation boundary is.Lab accuracy is measured by the Mean Lab Value (MLV) and Standard Deviation (SD) Lab accuracy is measured by Mean Lab Value and Standard Deviation. The higher the Mean Lab Value is, the clearer the color extraction effect is; the smaller the Standard Deviation is, the more concentrated the color distribution is in the color region, and the better the segmentation stability is. The formula is shown in equation (16).

$$\mu = \frac{1}{n}\sum_{i=1}^{n} x_i \quad \sigma = \sqrt{\frac{1}{n}\sum_{i=1}^{n}(x_i - \mu)^2} \tag{16}$$

In Eq. (16) $x_i$ denotes the color value of the $i$th pixel point; $\mu$ denotes the average color value; $\sigma$ denotes the standard deviation of the color value; and $n$ denotes the number of pixels.

**4.1.2. Performance test of color segmentation model for printed fabrics combining adaptive neural network and density peak clustering algorithm.** The study uses the color fabric sample dataset Version1 as the experimental dataset, which samples are organized and open-sourced by Hongwei Zhang Artificial Intelligence Laboratory, School of Electronic Information, Xi'an University of Engineering. The dataset contains a total of 3542 sample images, covering 17 different types of printed fabrics, with a uniform image resolution of 512×512×3, with strong representativeness and a wide range of texture and color distribution features. The study divides this dataset into a training set and a test set in the ratio of 8:2, i.e., about 2833 images are used for model training, and the remaining 709 images are used for model testing. No additional external samples are introduced to the test set, and all data are taken from the part of the Version1 dataset that is not involved in training.

In addition, in order to further evaluate the adaptability of the model in actual printing scenarios, four types of commercially available cotton printed fabric samples were additionally collected, uniformly cropped to 20 cm×20 cm, with resolutions of 150×150 and 300×300, respectively, and processed in a standard light box for image acquisition and testing, which were used to assist in evaluating the model's color recognition under different pixel densities. Stability and robustness. In order to improve the stability and generalization ability of the results, the study adopted a five-fold cross-validation approach to evaluate the performance of the model. The results are shown in Table 1.

From Table 1, the performance of the model is relatively stable in different folds: the ΔE color difference index fluctuates between 0.68 and 0.72, with an average of 0.71, which indicates that the color segmentation error is relatively small; the contour coefficient SC averages 0.85, which reflects the model's good compactness and differentiation of the clustering structure; the average execution time is 42.5 seconds, which demonstrates high processing efficiency; and the Lab detection accuracy is stable, ranging from 86.94% to 87.51%, with an average of 87.02%. between 86.94% and 87.51%, with an average of 87.02%. Compared with the single partition evaluation, the cross-validation further verifies the robustness and generalization ability of the model, reduces the variance effect of data partitioning, and ensures the reliability of the results. The execution time of traditional SOM neural network, traditional DPC algorithm, Fuzzy C-Means Clustering (FCM) and the research proposed SOM-DPC algorithm are tested respectively. As one of the classic image color segmentation algorithms, FCM has good clustering adaptability and a wide application base, and can deal with color regions with fuzzy boundaries more effectively, so it is representative in the color clustering task of fabric images. Fig 7 displays the experimental test results.

Fig 7a shows the line graph of the execution time of the four algorithms in the training set and Fig 7b displays the line graph of the execution time of the four algorithms in the test set. From Fig 7, in overall, the data in the test set performs better than the training set. Among them, the execution time of SPM-DPC algorithm for color segmentation in the training set increases gradually with the increase of the printed fabric samples, and the longest time can be up to 60 s. In the test set, the execution time of SPM-DPC algorithm is significantly reduced after training, and the longest processing time is

**Table 1. Five-fold cross-validation result.**

| Fold | Aberration | SC | Execution time (s) | Lab detection accuracy |
|---|---|---|---|---|
| Fold 1 | 0.71 | 0.84 | 42.56 | 86.93 |
| Fold 2 | 0.68 | 0.87 | 43.17 | 87.51 |
| Fold 3 | 0.73 | 0.83 | 41.91 | 86.47 |
| Fold 4 | 0.69 | 0.85 | 42.04 | 87.02 |
| Fold 5 | 0.72 | 0.86 | 42.72 | 87.16 |
| Average value | 0.71 | 0.85 | 42.44 | 87.02 |

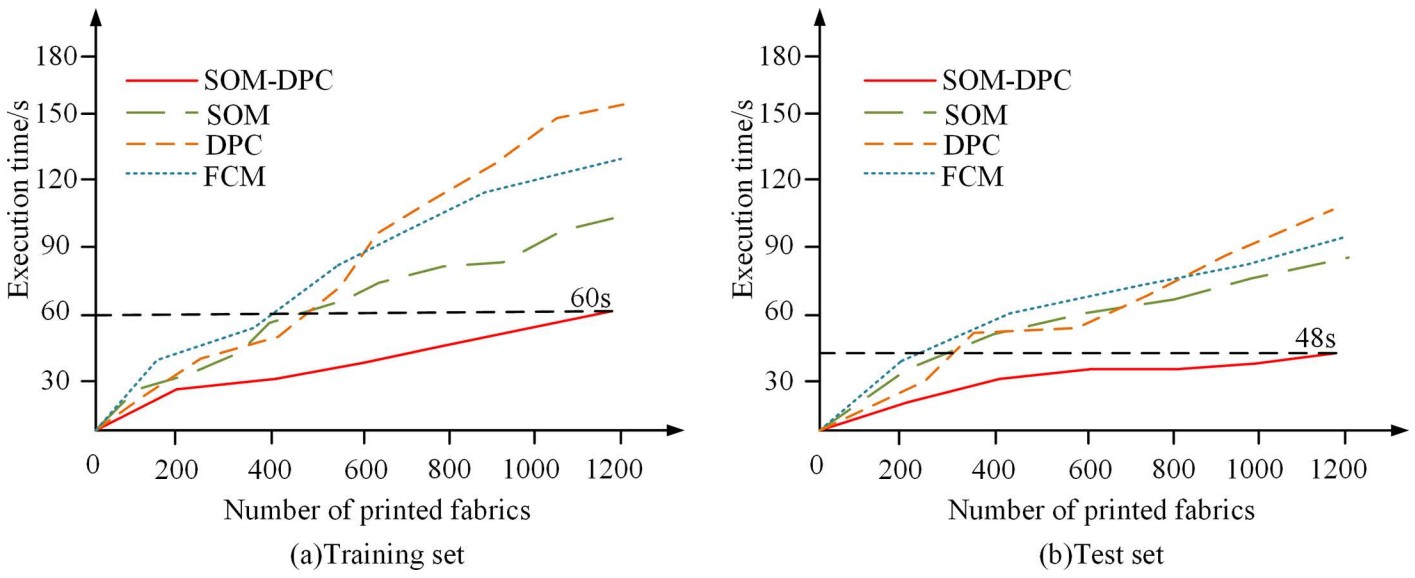

**Fig 7. Comparison of Execution Times of Different Algorithms.**

48 s. Compared with the SOM algorithm, the execution time of SPM-DPC algorithm is shortened by nearly 40%. In summary, the SPM-DPC algorithm proposed in the study is the fastest and more time-saving in performing the color segmentation task.

After comparing the execution speed of the color segmentation task, four printed fabrics were randomly selected as test objects for automatic color segmentation and extraction. The study continues to compare these four algorithms with the color difference reference index. Use the color difference formula to calculate the average color of each region, and use Datacolor SF 600 to measure the color difference between the colors of each region, the smaller the value of the color difference, indicating that the color segmentation is more reasonable and clear. Finally, statistically analyze the execution time of each color differentiation algorithm from determining the best color to segmenting each color region of the printed fabric. The specific color difference value test results are shown in Fig 8.

In Fig 8, after comparing the color segmentation results of the four algorithms, it is found. After the general algorithm color segmentation, the color difference values of each region are above 1.0. And from the color segmentation results of the SOM-DPC, for the four kinds of printed fabrics after the color differentiation, the color chromatic aberration of each region averages around 0.7, which is significantly lower compared to the color aberration values of the other three algorithms. It can be shown that the SOM-DPC can accurately experiment the color segmentation of printed fabrics, which makes the color measurement of each region more accurate.

To improve the accuracy of the color difference data obtained from the test, the study continues to take the average color value and standard deviation as the reference index, and analyze the data of the color difference experiment in detail. The average color value ranges from 0 to 100, and the larger the value, the more obvious the color distinction is. The standard deviation ranges from 0–10, the smaller the value, the better the segmentation effect. More advanced algorithms are also introduced for comparison, such as Graph Attention-based Segmentation Algorithm (GASA) and Transfer Learning-based Hyperspectral Color Segmentation (TL-HCS). Hyperspectral Color Segmentation, TL-HCS). The specific test results are displayed in Table 2.

From Table 2, among the detection results of the four algorithms, the SOM-DPC algorithm generally has a high average color value for the detection of the four kinds of printed fabrics, with the highest value reaching 87.49. For the comparison

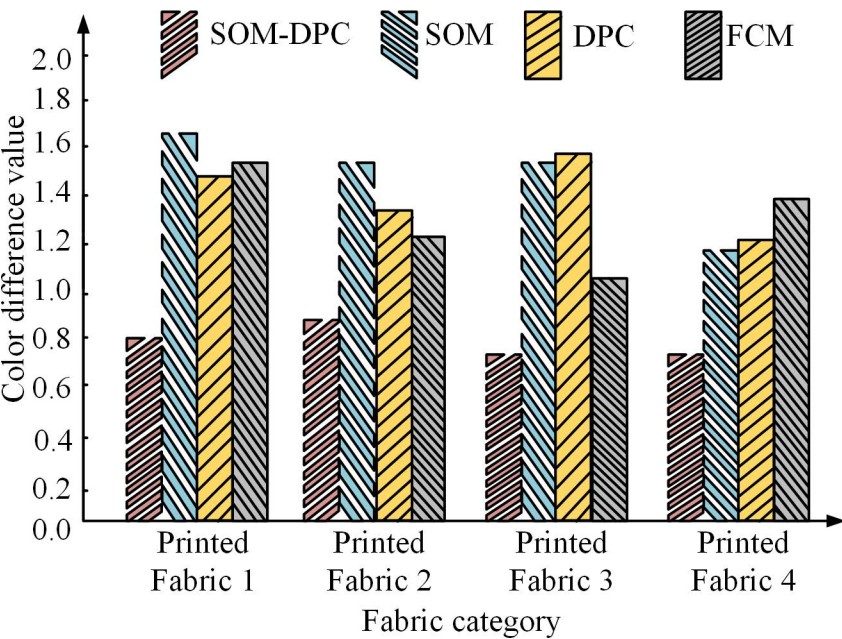

**Fig 8. Comparison of Color Differences in Color Regions after Segmentation by Different Algorithms.**

**Table 2. Average color values and standard deviation of different algorithms.**

| Variable | Average color value | | | | | | Standard deviation | | | | | |
|---|---|---|---|---|---|---|---|---|---|---|---|---|
| / | SOM-DPC | SOM | DPC | FCM | GASA | TL-HCS | SOM-DPC | SOM | DPC | FCM | GASA | TL-HCS |
| Printed Fabric 1 | 80.21 | 64.12 | 77.38 | 77.98 | 79.67 | 83.02 | 3.11 | 5.56 | 4.02 | 4.83 | 3.56 | 2.75 |
| Printed Fabric 2 | 87.49 | 54.96 | 67.51 | 78.14 | 81.55 | 86.73 | 2.18 | 5.36 | 4.89 | 4.38 | 2.83 | 2.34 |
| Printed Fabric 3 | 81.35 | 67.52 | 64.39 | 69.59 | 80.83 | 84.02 | 2.58 | 7.86 | 6.71 | 6.52 | 3.41 | 2.63 |
| Printed Fabric 4 | 79.83 | 59.86 | 68.94 | 65.74 | 77.62 | 81.94 | 2.97 | 5.88 | 6.98 | 4.67 | 3.33 | 2.58 |

of the standard deviation of the detection of the four kinds of printed fabrics, the SOM-DPC has the smallest value of the standard deviation, with the smallest value of up to 2.18. In summary, in the color segmentation test of different printed fabrics, the SOM-DPC proposed in the study has the best overall performance and the best performance.

**4.1.3. Application test of color segmentation model for printed fabrics combining adaptive neural network and density peak clustering algorithm.** In order to test the performance status of the SOM-DPC segmentation algorithm in a more in-depth manner, four commercially available cotton printed fabrics were selected for the study. The main material is cotton, and the pattern styles cover geometric, floral and gradient types to enhance the adaptability of the algorithm under diverse pattern structures. The fabric size is specified as 20cmx20 cm, and the specimen fabrics are placed in the standard light source box of BZGY908A, and the light source irradiates the fabric surface at an angle of 45 degrees. After the acquisition of the respective hyperspectral images, the printed fabrics with two sets of pixels are selected for comparison testing through ENVI4.8. The printed fabrics used for experimental testing are shown in Fig 9.

After determining the experimental materials, the four groups of printed fabrics were divided into two categories, one with 150x150 pixels and the other with 300x300 pixels for 8 kinds of printed fabrics. The actual color segmentation of SOM-DPC segmentation algorithm is tested using Lab value as a reference index. Where L indicates the brightness of the color, a positive number indicates whitish, and a negative number indicates blackish. a indicates the

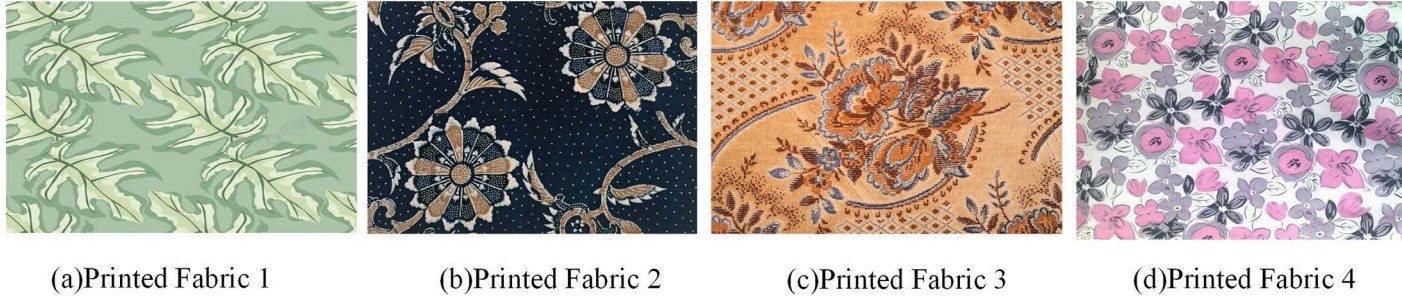

(a)Printed Fabric 1　　　　(b)Printed Fabric 2　　　　(c)Printed Fabric 3　　　　(d)Printed Fabric 4

**Fig 9. Four types of printed fabrics for experimental testing.**

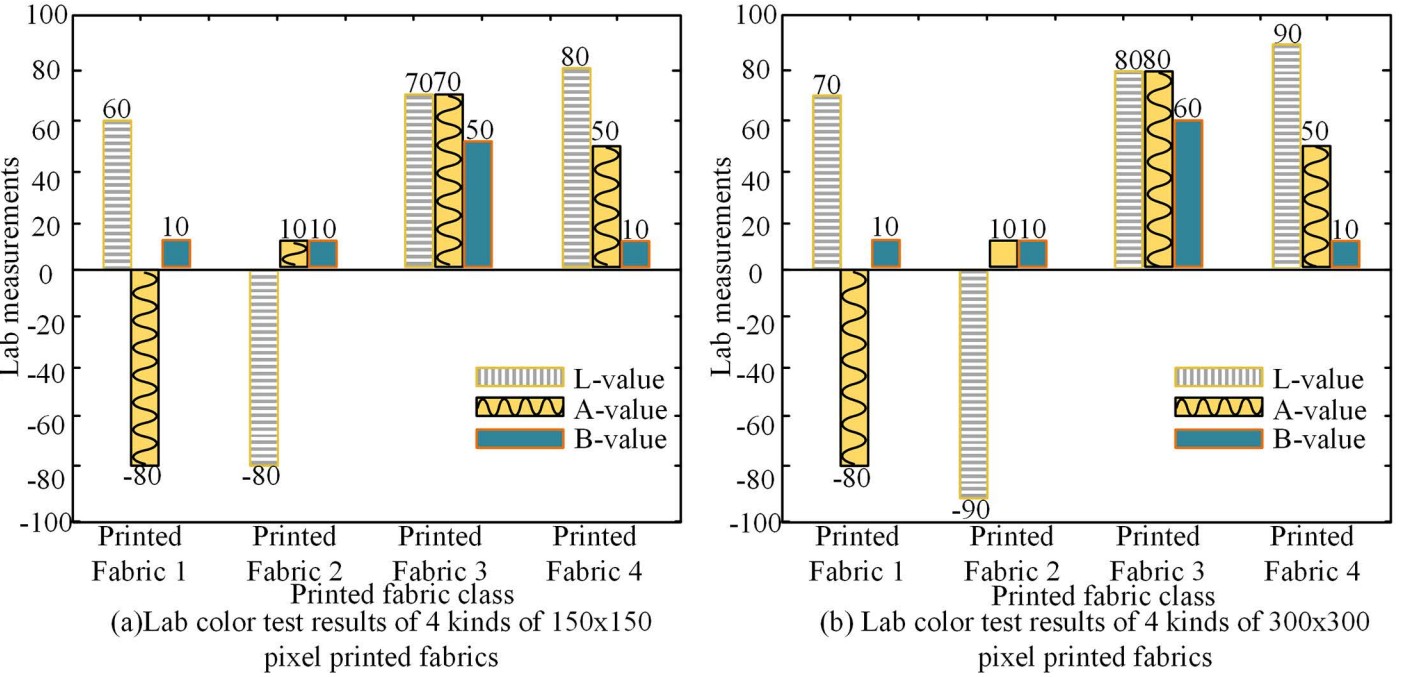

(a)Lab color test results of 4 kinds of 150x150 pixel printed fabrics

(b) Lab color test results of 4 kinds of 300x300 pixel printed fabrics

**Fig 10. Lab test results of printed fabrics with different pixels.**

red-green value, a positive number indicates reddish, and a negative number indicates greenish. b indicates the yellow-blue value, a positive number indicates yellowish, and a negative number indicates bluish. Specific experimental measurements are shown in Fig 10.

Fig 10a shows the Lab detection results of SOM-DPC segmentation algorithm for four printed fabrics with 150x150 pixels and Fig 10b shows the Lab detection results of SOM-DPC segmentation algorithm for four printed fabrics with 300x300 pixels. From Fig 10, the detection results given visually do not differ much from the real ones. The accuracy of Lab detection increases with the increase in pixels. However, visually fabric 4 should not have yellow color appearing, while 10 points of yellow color detection value existed in the inspection results. Therefore, in order to show the visual segmentation effect of the SOM-DPC algorithm in a more graphic way, the study converted the spectral reflectance of each pixel point within the hyperspectral image into Lab plots, and the fabric 1

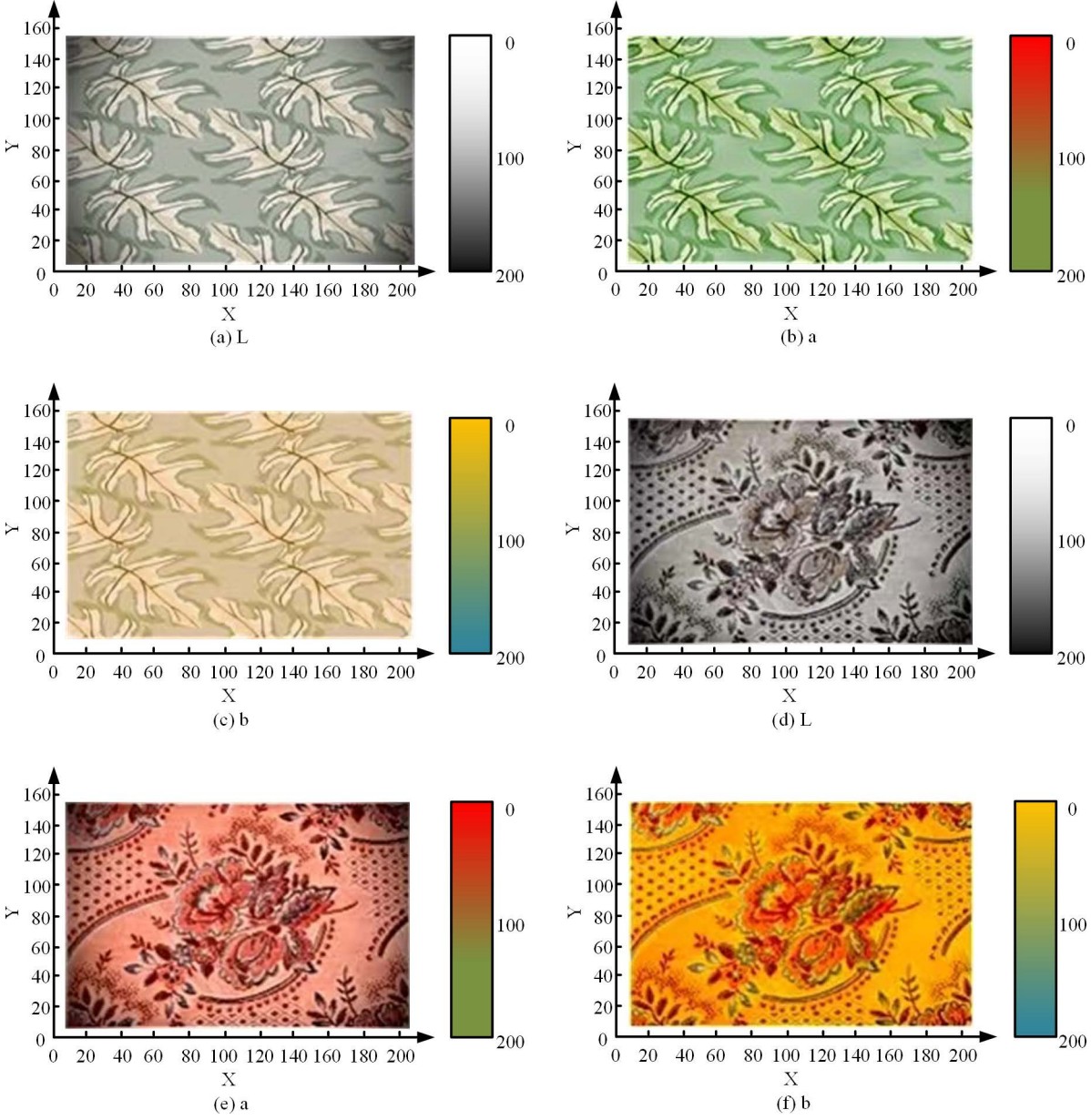

**Fig 11. Lab visualization results for two fabrics.**

and fabric 3, which have smaller segmentation differences, were visualized and tested, and the results of the test are shown in Fig 11.

Fig 11a, b, and c are Lab test charts for fabric 1, and Fig 11d, e, and f are Lab test charts for fabric 3. As can be seen in Fig 11, the results of the Lab value performance of Fabric 1 are generally consistent with the test results in Fig 10, exhibiting stronger L and a values, with fewer b values, i.e., more white and green, with a small amount of yellow. Fabric 3 exhibited stronger Lab values, i.e., more white and red, with some amount of green. Taken together, the above results reveal that the SOM-DPC model can be adapted to the color segmentation recognition of diverse printed fabrics, and its test results have a certain degree of realism.

## 5. Discussion

A color segmentation method for printed fabrics integrating SOM and DPC algorithms is proposed, aiming to improve the color segmentation accuracy and processing efficiency under complex textures. By introducing the adaptive topology mapping capability of SOM for downscaling the representation of high-dimensional image features and combining it with the density-oriented clustering strategy of DPC, the fine segmentation of the main color region of printed fabrics is achieved. The experimental results show that the SOM-DPC algorithm significantly outperforms the traditional method in a number of indexes, such as color difference control, average color value, standard deviation and contour coefficient, and exhibits stronger segmentation stability and region identification ability. Compared with the existing studies, the traditional SOM has limited recognition ability in the face of similar color values or blurred regions, DPC has the ability of non-spherical clustering, but there is a risk of unstable peak recognition in the direct clustering of high-dimensional images, and FCM, although it is widely used in color segmentation, is easily affected by the initial value setting and noise interference, and there are limitations in the color continuity and the accuracy of small-region segmentation. Compared with Rodrigues et al. [20] 's fault diagnosis study based on the combination of DPC and dynamic neural network, the method in this paper pays more attention to the synergistic optimization of the color clustering structure and topology mapping, which can effectively deal with the problem of nonlinear color distribution. Meanwhile, unlike the study of Lei et al. [21] which emphasizes the efficiency of local stratification of density peaks, the present method pays more attention to the global consistency and local boundary preservation of the main color region in the print image. In addition, the segmentation process of the SOM-DPC algorithm does not require a large number of a priori parameter settings, which provides strong adaptability and practical generalization. Nevertheless, it is worth noting that, although the dataset used in the study covers a variety of printed fabric samples, the overall sample size is still limited, especially in the practical application, there may be more complex materials and color variation patterns that have not yet been covered. Therefore, although the experimental results are representative, the generalization ability in a wide range of scenarios still needs to be further verified. In the future, we will consider extending the model to larger multi-material fabric image datasets and introducing a cross-data source validation mechanism to enhance the generalizability and practical value of the model.

## Author contributions

**Writing – original draft:** Niu Meng.

**Writing – review & editing:** Niu Meng.

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
