## [Decision Letter · Decision Letter 0]

Dear Dr. Meng,

Thank you for submitting your manuscript to PLOS ONE. After careful consideration, we feel that it has merit but does not fully meet PLOS ONE’s publication criteria as it currently stands. Therefore, we invite you to submit a revised version of the manuscript that addresses the points raised during the review process.

We look forward to receiving your revised manuscript.

Kind regards,

Jeevithan Elango, PhD

Academic Editor

PLOS ONE

 [Ministry of Education Humanities and Social Sciences Research Youth Fund Project: A Study of the Military Attire Formations in Southern Inspection Paintings of the Ming and Qing Dynasties.]. 

6. In the online submission form, you indicated that [The datasets used and/or analysed during the current study available from the corresponding author on reasonable request.].

Additional Editor Comments:

Major Revision

Reviewers' comments:

Reviewer's Responses to Questions

**Comments to the Author**

1. Is the manuscript technically sound, and do the data support the conclusions?

Reviewer #1: Yes

Reviewer #2: Yes

2. Has the statistical analysis been performed appropriately and rigorously?

Reviewer #1: No

Reviewer #2: Yes

3. Have the authors made all data underlying the findings in their manuscript fully available?

Reviewer #1: Yes

Reviewer #2: No

4. Is the manuscript presented in an intelligible fashion and written in standard English?

Reviewer #1: No

Reviewer #2: Yes

Reviewer #1: Abstract

color segmentation reaches 0.7, 0.7 is what evaluation index? Same as 87.49, 2.18

The motivation of this paper is not clear, existing segmentation methods has two problems, are you sure? It means your proposed algorithm is the best one.

Need to report compare with which algorithms and the results.

Introduction

It is too short, adding the motivation, how to solve the problems, the paper’s contributions

3.1.2 Printed fabric color segmentation model construction by combining DPC

algorithm and

The title is wrong

Compare the proposed method with new algorithms

Invited native speaker to improve the paper.

Reviewer #2: 1. Literature Review and Contribution Clarity:

1.1 The novelty claim regarding the integration of SOM and DPC lacks sufficient contextual support. The paper states, "There are few examples of combining adaptive neural networks and density peak clustering algorithms for color segmentation of printed fabrics." However, the motivation for this work remains unclear, particularly given that previous related works have combined SOM and DPC. The paper should discuss the results and limitations of these earlier studies and clearly demonstrate how the proposed method addresses these issues.

1.2 The literature review summarizes prior works but does not sufficiently engage in a critical analysis of their limitations. It fails to clearly identify the specific knowledge gaps this paper addresses. I recommend expanding the literature review to highlight these gaps explicitly and position this work's contribution more clearly in the context of existing research.

1.3 To better position this work relative to the state-of-the-art, I would recommend citing more recent (2023-2024) and high-impact studies in the fields of image segmentation, adaptive clustering, and fabric analysis.

1.4 The manuscript sometimes introduces citations (e.g., "Son N N et al.") in an awkward and grammatically incorrect manner. I recommend integrating references more naturally into the sentence structure, such as "Son et al. [5] proposed..." to maintain academic tone and improve readability.

1.5 Many concluding sentences (e.g., "the practical application is better and the functional performance has effectiveness") are vague and grammatically awkward. I recommend rephrasing such conclusions using clear academic expressions, maintaining a formal style.

2. Methods

2.1 The dataset information is insufficient and inconsistently presented. It remains unclear whether the 20% test split originated solely from the AITEX dataset or if it includes the additional printed fabric samples. I recommend adding a dedicated 'Dataset Description' section and clearly describing the train/test split and the characteristics of all datasets used.

2.2 Evaluation metrics are mentioned but not systematically introduced or justified. A standalone 'Evaluation Metrics' section defining color difference (ΔE), Silhouette Coefficient (SC), execution time, and Lab accuracy should be added for clarity.

2.3 The experimental dataset is small compared to related works, limiting the generalizability of the results. I suggest discussing the dataset limitation to improve the paper's credibility.

2.4 Given the small size of the experimental dataset, I strongly recommend applying k-fold cross-validation to ensure the generalizability and reliability of the reported results. Alternatively, if cross-validation is not feasible, the manuscript should clearly discuss the rationale for using a simple 80/20 split and address the potential risks of variance and overfitting.

2.5 Section 3.1.2 title is incomplete and awkwardly phrased ("combining DPC algorithm and..."). I recommend revising it for clarity, completeness, and consistency with the terminology used elsewhere in the manuscript

3. Results

3.1 The manuscript introduces the abbreviation "FCM" without defining it upon first use. I recommend expanding what it stands for.

3.2 While comparing the proposed SOM-DPC method to SOM and DPC individually is logical and expected, the manuscript does not justify the choice of FCM as an additional comparison method. I recommend explaining why FCM was selected as a baseline, and possibly citing related works where FCM has been used in similar fabric color segmentation tasks.

3.3 Sections 4.1.1 and 4.1.2 are not clearly distinct. Section 4.1.1 partially contains methodological content that should be described earlier, such as the experiment environment, while Section 4.1.2 mixes methodological explanation with experimental results. I suggest removing methodological parts in 4.1.1 and focusing 4.1 entirely on experimental validation of SOM-DPC's performance, following a clear structure for reporting results.

4. Discussion

4.1 The Discussion section mainly repeats the results without deeper interpretation, theoretical reflection, or critical comparison to related studies. I recommend substantially revising the Discussion to interpret the findings in depth, contextualize them within existing literature. A proper, standalone Conclusion section is missing. It is crucial to summarize key findings, contributions, limitations, and future work prospects.

4.2 There is an inconsistency regarding the number and types of fabrics used. While five fabrics are mentioned in the experiments section, the Discussion implies a single material type (cotton). The manuscript should clarify the fabric types used (material composition, pattern characteristics) and ensure consistency throughout the text.

5.Writing Quality

Revise the manuscript for language, grammar, and academic tone.

**Do you want your identity to be public for this peer review?** For information about this choice, including consent withdrawal, please see our Privacy Policy

Reviewer #1: No

Reviewer #2: No

---

## [Author Response · Author response to Decision Letter 1]

9 Jun 2025

Review Comments to the Author

Reviewer #1:

1.Abstract

color segmentation reaches 0.7, 0.7 is what evaluation index? Same as 87.49, 2.18

The motivation of this paper is not clear, existing segmentation methods has two problems, are you sure? It means your proposed algorithm is the best one.

Need to report compare with which algorithms and the results.

Reply: Thank you for your comments. 0.7 is the average color difference evaluation index, which is not the same as 87.49 and 2.18. Meanwhile, the motivation of the study was rephrased, while the description of the results was optimized, the comparison of other methods was added, and the modifications were as follows:

With the development of computer vision and image processing technology, color segmentation of printed fabrics has gradually become a key task in the textile industry. However, the existing methods often face the problems of low segmentation accuracy and poor computational efficiency when dealing with high complexity patterns and similar colors. To address the above problems, a new color segmentation algorithm for printed fabrics is proposed by integrating the self-organizing mapping network (SOM) in adaptive neural network and the density peak clustering algorithm. The method achieves topological mapping learning of color features through SOM, and then uses DPC for density-driven fine clustering division, which effectively improves the accuracy and stability of color segmentation. The experimental results show that the proposed method shortens the execution time by nearly 40% compared with the self-organized mapping network, and the average color difference (ΔE) of each region after color segmentation is as low as 0.7, which is significantly better than other algorithms. Meanwhile, in the detection of the four types of printed fabric samples, the obtained average color value is up to 87.49 (the higher the 0-100 score value indicates that the color is more significant), and the smallest standard deviation is 2.18 (the smaller the value indicates that the color segmentation is more centralized), which further verifies the comprehensive advantages of the algorithm in terms of segmentation accuracy and stability. In conclusion, the proposed method provides an effective reference for improving the quality and efficiency of color segmentation of printed fabrics.

2.Introduction

It is too short, adding the motivation, how to solve the problems, the paper’s contributions

Reply: Thank you for your comments. The introductory section of the manuscript has been revised to add a description of motivation, problem solution, and contributions, as modified below:

Motivation:

Although both algorithms have their own characteristics, SOM suffers from insufficient boundary refinement and sensitivity to local structure distribution, while DPC faces the challenges of easy centroid misalignment and noise misjudgment in high-dimensional space. Based on this, the study proposes a color segmentation algorithm for printed fabrics that integrates SOM and DPC, aiming to further improve the image segmentation effect of printed fabrics.

Solution:

Specifically, on the one hand, the adaptive mapping and region delineation of high-dimensional color features are accomplished by the topology preservation ability of self-organizing mapping network in unsupervised learning, and on the other hand, the density peak clustering is used to enhance the density and cluster center determination of the initial results of the SOM, so as to improve the differentiation ability of the complex color structure and the boundary integrity expression.

Contribution:

The contribution of the study is that a color segmentation model with both global mapping and local aggregation is proposed, which systematically solves the problems of poor identification of color proximity regions, weak expression of pattern boundaries and strong dependence on parameter tuning in the traditional methods, and provides theoretical support and methodological basis for improving the stability, scalability and industrial practicability of image segmentation of complex textured fabrics.

3.

3.1.2 Printed fabric color segmentation model construction by combining DPC algorithm and

The title is wrong

Reply: Thank you for your comments. The title of section 3.1.2 of the manuscript has been revised as follows:

3.1.2 Color Segmentation Model Construction of Printed Fabrics Combined with DPC Algorithm

4.Compare the proposed method with new algorithms

Reply: Thank you for your comments. Table 2 (formerly Table 1) of the manuscript has been modified by adding new algorithms for the modifications, which are as follows:

More advanced algorithms are also introduced for comparison, such as Graph Attention-based Segmentation Algorithm (GASA) and Transfer Learning-based Hyperspectral Color Segmentation (TL-HCS). Hyperspectral Color Segmentation, TL-HCS). The specific test results are displayed in Table 2.

Tab 2 Average color values and standard deviation of different algorithms

Variable Average color value Standard deviation

/ SOM-DPC SOM DPC FCM GASA TL-HCS SOM-DPC SOM DPC FCM GASA TL-HCS

Printed Fabric 1 80.21 64.12 77.38 77.98 79.67 83.02 3.11 5.56 4.02 4.83 3.56 2.75

Printed Fabric 2 87.49 54.96 67.51 78.14 81.55 86.73 2.18 5.36 4.89 4.38 2.83 2.34

Printed Fabric 3 81.35 67.52 64.39 69.59 80.83 84.02 2.58 7.86 6.71 6.52 3.41 2.63

Printed Fabric 4 79.83 59.86 68.94 65.74 77.62 81.94 2.97 5.88 6.98 4.67 3.33 2.58

5. Invited native speaker to improve the paper.

Reply: Thank you for your comments. Statements throughout the manuscript have been touched up to improve readability and flow.

Reviewer #2:

1. Literature Review and Contribution Clarity:

1.1 The novelty claim regarding the integration of SOM and DPC lacks sufficient contextual support. The paper states, "There are few examples of combining adaptive neural networks and density peak clustering algorithms for color segmentation of printed fabrics." However, the motivation for this work remains unclear, particularly given that previous related works have combined SOM and DPC. The paper should discuss the results and limitations of these earlier studies and clearly demonstrate how the proposed method addresses these issues.

Reply: Thank you for your comments. Section 2 of the manuscript, Related works section, has been revised to re-explain the results and limitations of the earlier study and to clearly state the solution of the proposed methodology, as modified below:

In summary, although the combination of SOM and clustering methods has been applied in some literatures, such as realizing clustering in remote sensing images or texture images, most of these methods have not considered the consistency requirements of the high-dimensional spatial distribution of the color features in the printed fabrics with the expression of the physical chromatic aberration, and they have not integrated the advantages of the self-organized mapping of the neural network and the nonparametric clustering ability of the DPC. Therefore, this study focuses on the integration of SOM and DPC, and proposes a SOM-DPC color segmentation model for printed fabrics by introducing a clustering mechanism based on density distribution while maintaining the advantages of spatial topological mapping structure. The model not only improves the accuracy of color recognition in hyperspectral images, but also enhances the sensitivity to the adjacent color gamut demarcation, thus filling in the deficiencies of existing methods in the control of high-dimensional color representation and segmentation accuracy.

1.2 The literature review summarizes prior works but does not sufficiently engage in a critical analysis of their limitations. It fails to clearly identify the specific knowledge gaps this paper addresses. I recommend expanding the literature review to highlight these gaps explicitly and position this work's contribution more clearly in the context of existing research.

Reply: Thank you for your comments. Section 2 of the manuscript, Related works section, has been revised to critically analyze each piece of literature and point out gaps in the research, as modified below:

Shi et al. [9] combined DPC with a probabilistic neural network for analog circuit fault diagnosis to improve classification accuracy while reducing the number of neurons. Guan et al. [10] optimized the hierarchical clustering efficiency of DPC on large-scale datasets through a correlation transfer mechanism. While Li et al. [11] combined DPC with an RBF network for mixed-data classification and achieved a high accuracy of 97.52%. In addition, Zheng et al. [12] optimized DPC based on k-nearest neighbor for industrial process monitoring and significantly improved detection stability. Although DPC shows good performance in the field of image analysis, its clustering results are sensitive to local density estimation and distance metrics, which makes it difficult to directly capture complex color structures in high-dimensional images. Some studies have attempted to integrate neural networks with clustering algorithms. For example, Gharehchopogh et al. [13] improved the image segmentation performance by improving the African vulture optimization algorithm, but its applicable images are mainly medical and natural images; Zhou et al. [14] improved the accuracy and speed of fabric defect detection based on the improved YOLOv5s model, but the target is more oriented to structural defect recognition; Sikka et al. [15] constructed a fabric detection model based on multilevel backpropagation network to construct a fabric detection model, which is difficult to cope with the problem of boundary extraction in the region of similar color despite its robustness. There is a lack of research on color adaptive recognition and high-resolution segmentation for high-density, multi-color fusion scenarios in printed fabrics.

1.3 To better position this work relative to the state-of-the-art, I would recommend citing more recent (2023-2024) and high-impact studies in the fields of image segmentation, adaptive clustering, and fabric analysis.

Reply: Thank you for your comments. Section 2 of the manuscript, Related works, has been revised to add the latest relevant literature for the years 2023-2025, as modified below:

Some studies have attempted to integrate neural networks with clustering algorithms. For example, Gharehchopogh et al. [13] improved the image segmentation performance by improving the African vulture optimization algorithm, but its applicable images are mainly medical and natural images; Zhou et al. [14] improved the accuracy and speed of fabric defect detection based on the improved YOLOv5s model, but the target is more oriented to structural defect recognition; Sikka et al. [15] constructed a fabric detection model based on multilevel backpropagation network to construct a fabric detection model, which is difficult to cope with the problem of boundary extraction in the region of similar color despite its robustness. There is a lack of research on color adaptive recognition and high-resolution segmentation for high-density, multi-color fusion scenarios in printed fabrics.

[13] Gharehchopogh F S, Ibrikci T. An improved African vultures optimization algorithm using different fitness functions for multi-level thresholding image segmentation. Multimedia Tools and Applications, 2024, 83(6): 16929-16975.

[14] Zhou S, Zhao J, Shi Y S, Wang F Y, Mei S Q. Research on improving YOLOv5s algorithm for fabric defect detection. International Journal of Clothing Science and Technology, 2023, 35(1): 88-106.

[15] Sikka M P, Sarkar A, Garg S. Artificial intelligence (AI) in textile industry operational modernization. Research Journal of Textile and Apparel, 2024, 28(1): 67-83.

1.4 The manuscript sometimes introduces citations (e.g., "Son N N et al.") in an awkward and grammatically incorrect manner. I recommend integrating references more naturally into the sentence structure, such as "Son et al. [5] proposed..." to maintain academic tone and improve readability.

Reply: Thank you for your comments. Changes have been made to section 2 of the manuscript, Related works, to adjust the way the literature is cited as you suggested, and an example of the changes is shown below:

Son et al. [5] used ANN for piezoelectric ceramic actuator control, which significantly improves tracking accuracy and stability, but its application scope is concentrated in the dynamic control field.

1.5 Many concluding sentences (e.g., "the practical application is better and the functional performance has effectiveness") are vague and grammatically awkward. I recommend rephrasing such conclusions using clear academic expressions, maintaining a formal style.

Reply: Thank you for your comments. Some of the phrases you mentioned have been revised, and the phrases throughout the manuscript have been touched up to improve readability and flow. The changes are as follows:

Shi et al. [9] combined DPC with a probabilistic neural network for analog circuit fault diagnosis to improve classification accuracy while reducing the number of neurons. Guan et al. [10] optimized the hierarchical clustering efficiency of DPC on large-scale datasets through a correlation transfer mechanism. While Li et al. [11] combined DPC with an RBF network for mixed-data classification and achieved a high accuracy of 97.52%.

2. Methods

2.1 The dataset information is insufficient and inconsistently presented. It remains unclear whether the 20% test split originated solely from the AITEX dataset or if it includes the additional printed fabric samples. I recommend adding a dedicated 'Dataset Description' section and clearly describing the train/test split and the characteristics of all datasets used.

Reply: Thank you for your comments. Section 4 of the manuscript has been revised to address the uniform additions and optimization of the dataset descriptions as follows:

The study uses the color fabric sample dataset Version1 as the experimental dataset, which samples are organized and open-sourced by Hongwei Zhang Artificial Intelligence Laboratory, School of Electronic Information, Xi'an University of Engineering. The dataset contains a total of 3542 sample images, covering 17 different types of printed fabrics, with a uniform image resolution of 512×512×3, with strong representativeness and a wide range of texture and color distribution features. The study divides this dataset into a training set and a test set in the ratio of 8:2, i.e., about 2833 images are used for model training, and the remaining 709 images are used for model testing. No additional external samples are introduced to the test set, and all data are taken from the part of the Version1 dataset that is not involved in training.

2.2 Evaluation metrics are mentioned but not systematically introduced or justified. A standalone 'Evaluation Metrics' section defining color difference (ΔE), Silhouette Coefficient (SC), execution time, and Lab accuracy should be added for clarity.

Reply: Thank you for your comments. Section 4 of the manuscript has been revised to create a new section 4.1.1, the Assessment Indicators section, as modified below:

4.1.1 Assessment of indicators

In order to systematically measure the effectiveness of the proposed SOM-DPC color segmentation algorithm for printed fabrics, the study employs four types of common evaluation metrics for performance evaluation, which are Color Difference, Silhouette Coefficient (SC), Execution Time, and Lab Accuracy metrics. Color difference is used to measure the color distance between different color regions, which is calculated based on CIE Lab color space and defined as shown in equation (14).

(14)

In Eq. (14), , , and , , represent the luminance (L), red-green axis (a) and yellow-blue axis (b) coordinate values of two color regions in Lab space, respectively. The smaller value of color difference represents the more accurate color segmentation and the more natural color transition of adjacent regions. The contour coefficient is used to measure the tightness of each data point in the clustering result with its cluster and the separation of other

---

## [Decision Letter · Decision Letter 1]

Colour Segmentation of Printed Fabrics by Integrating Adaptive Neural Network and Density Peak Clustering Algorithm

PONE-D-25-06507R1

Dear Dr. Meng,

We’re pleased to inform you that your manuscript has been judged scientifically suitable for publication and will be formally accepted for publication once it meets all outstanding technical requirements.

Kind regards,

Jeevithan Elango, PhD

Academic Editor

PLOS ONE

Additional Editor Comments (optional):

No more comments.

Reviewers' comments:

Reviewer's Responses to Questions

**Comments to the Author**

Reviewer #2: All comments have been addressed

2. Is the manuscript technically sound, and do the data support the conclusions?

Reviewer #2: Yes

3. Has the statistical analysis been performed appropriately and rigorously?

Reviewer #2: Yes

4. Have the authors made all data underlying the findings in their manuscript fully available?

Reviewer #2: Yes

5. Is the manuscript presented in an intelligible fashion and written in standard English?

Reviewer #2: Yes

Reviewer #2: The author has adequately addressed all the comments raised in a previous round of review and I feel that this manuscript is now acceptable for publication. The Literature Review and Contribution Clarity, Methods, Results and Discussion sections have been revised.

**Do you want your identity to be public for this peer review?** For information about this choice, including consent withdrawal, please see our Privacy Policy

Reviewer #2: No

---

## [Editor Report · Acceptance letter]

PONE-D-25-06507R1

PLOS ONE

Dear Dr. Meng,

I'm pleased to inform you that your manuscript has been deemed suitable for publication in PLOS ONE. Congratulations! Your manuscript is now being handed over to our production team.

Kind regards,

on behalf of

Dr. Jeevithan Elango

Academic Editor

PLOS ONE